# Systematic Review and Modelling of Age-Dependent Prevalence of *Toxoplasma gondii* in Livestock, Wildlife and Felids in Europe

**DOI:** 10.3390/pathogens12010097

**Published:** 2023-01-06

**Authors:** Filip Dámek, Arno Swart, Helga Waap, Pikka Jokelainen, Delphine Le Roux, Gunita Deksne, Huifang Deng, Gereon Schares, Anna Lundén, Gema Álvarez-García, Martha Betson, Rebecca K. Davidson, Adriana Györke, Daniela Antolová, Zuzana Hurníková, Henk J. Wisselink, Jacek Sroka, Joke W. B. van der Giessen, Radu Blaga, Marieke Opsteegh

**Affiliations:** 1Anses, INRAE, Ecole Nationale Vétérinaire d’Alfort, Laboratoire de Santé Animale, BIPAR, F-94700 Maisons-Alfort, France; 2Centre for Infectious Disease Control—Zoonoses and Environmental Microbiology, National Institute for Public Health and the Environment, Antonie van Leeuwenhoeklaan 9, P.O. Box 1, 3720 BA Bilthoven, The Netherlands; 3Laboratório de Parasitologia, Instituto Nacional de Investigação Agrária e Veterinária, 2780-157 Oeiras, Portugal; 4Infectious Disease Preparedness, Statens Serum Institut, 2300 Copenhagen, Denmark; 5Institute of Food Safety, Animal Health and Environment BIOR, LV-1076 Riga, Latvia; 6Institute of Epidemiology, Friedrich-Loeffler-Institut, Federal Research Institute for Animal Health, 17493 Greifswald, Germany; 7Department of Microbiology, National Veterinary Institute, 75189 Uppsala, Sweden; 8SALUVET, Animal Health Department, Faculty of Veterinary Sciences, Complutense University of Madrid, 28040 Madrid, Spain; 9School of Veterinary Medicine, University of Surrey, Guildford GU2 7AL, UK; 10Food Safety and Animal Health, Norwegian Veterinary Institute, 9016 Tromsø, Norway; 11Department of Parasitology and Parasitic Diseases, Faculty of Veterinary Medicine, University of Agricultural Sciences and Veterinary Medicine Cluj-Napoca, 400372 Cluj-Napoca, Romania; 12Institute of Parasitology, Slovak Academy of Sciences, 040 01 Košice, Slovakia; 13Wageningen Bioveterinary Research, Wageningen University & Research, P.O. Box 65, 8200 AB Lelystad, The Netherlands; 14Department of Parasitology and Invasive Diseases, National Veterinary Research Institute, 24-100 Pulawy, Poland

**Keywords:** toxoplasmosis, pig, sheep, goat, cattle, cat, Bayesian model, systematic review, seroprevalence, meta-analysis

## Abstract

*Toxoplasma gondii* is a zoonotic parasite of importance to both human and animal health. The parasite has various transmission routes, and the meat of infected animals appears to be a major source of human infections in Europe. We aimed to estimate *T. gondii* prevalence in a selection of animal host species. A systematic literature review resulting in 226 eligible publications was carried out, and serological data were analyzed using an age-dependent Bayesian hierarchical model to obtain estimates for the regional *T. gondii* seroprevalence in livestock, wildlife, and felids. Prevalence estimates varied between species, regions, indoor/outdoor rearing, and types of detection methods applied. The lowest estimated seroprevalence was observed for indoor-kept lagomorphs at 4.8% (95% CI: 1.8–7.5%) and the highest for outdoor-kept sheep at 63.3% (95% CI: 53.0–79.3%). Overall, *T. gondii* seroprevalence estimates were highest within Eastern Europe, whilst being lowest in Northern Europe. Prevalence data based on direct detection methods were scarce and were not modelled but rather directly summarized by species. The outcomes of the meta-analysis can be used to extrapolate data to areas with a lack of data and provide valuable inputs for future source attribution approaches aiming to estimate the relative contribution of different sources of *T. gondii* human infection.

## 1. Introduction

*Toxoplasma gondii* is an important zoonotic protozoan parasite capable of infecting potentially all warm-blooded vertebrates [1]. Felids are the only known definitive hosts of *T. gondii* and a source of environmental oocyst contamination through shedding in feces [2,3,4]. Upon infection, *T. gondii* rapidly multiplies, enters the host’s tissues, and forms tissue cysts. There it remains infective, enabling *T. gondii* to reach a new host through the carnivory of its former host [5]. Infection is considered to be lifelong, and unspecific clinical signs may be observed. Most of the infections are, however, subclinical. *Toxoplasma gondii* can also be transmitted to the fetus, causing congenital toxoplasmosis and potentially resulting in a miscarriage or stillbirth [6]. The potential consequences of congenital toxoplasmosis in particular have contributed to the high ranking of *T. gondii* compared to other major food-borne parasites [7]. The global annual incidence of congenital toxoplasmosis was estimated to be 190,100 cases (95% CI: 179,300–206,300). This was equivalent to a burden of 1.20 million (95% CI: 0.76–1.90) disability-adjusted life years (DALYs) per annum [8]. Particularly high burdens were observed in South America and some Middle Eastern and low-income countries [8].

There are two main routes of acquired infection in humans: An environmental route through the ingestion of *T. gondii* oocysts present in contaminated water, soil, or fresh produce, and a meat-borne route through the consumption of tissue cysts that may be present in the meat of infected animals. The consumption of raw or undercooked meat of infected animals appears to be the main source of transmission of the parasite for humans in Europe [9]. The prevalence of *T. gondii* infections in the definitive hosts, felids, can give an indication of the environmental contamination with oocysts, and the surveillance of *T. gondii* prevalence within animal populations intended for human consumption helps to assess the risk of human infection from various meat products. Most livestock species have a herbivorous diet, therefore these animals mainly acquire the infection through ingesting sporulated *T. gondii* oocysts [10,11,12]. Wildlife may become infected by ingesting the meat of infected animals in addition to oocysts shed by felids. Stray cats and wild felids, in addition to domestic cats, contribute to soil contamination, possibly followed by runoff to surface waters [13,14].

Due to environmental contamination, the risk of acquiring *T. gondii* infection is higher in animals kept outdoors, thus leading to the difference in *T. gondii* prevalence between animals kept indoors or outdoors, as has been well-documented in various species [14,15,16]. In addition, prevalence is known to increase with age [14], in line with the assumption that the infection persists for the lifetime of the host.

Two types of methods are used to demonstrate *T. gondii* infection in animals. Direct methods such as a bioassay or PCR can be used to demonstrate the presence of the parasite or its DNA; however, indirect methods that demonstrate the presence of antibodies against *T. gondii* are more frequently used [17,18,19]. There appears to be a good correlation between the detection of antibodies and the presence of parasites in most animal species, thus seroprevalence provides an indication of the proportion of animals that presents a risk for human infection if consumed [20,21]. Moreover, since *T. gondii* parasites are clustered in sparsely distributed tissue cysts, and in case the sample used for direct detection methods is small, tissue cysts may remain undetected with direct detection methods. Therefore, the detection of anti-*T. gondii* specific immunoglobulin G (IgG) antibodies is considered the most sensitive indicator of *T. gondii* infection in most animal species. However, in cattle, buffaloes, and equids, the seroprevalence is known not to correspond well with the presence of infective cysts in the tissue of these animals [20,21,22]. For this reason, direct detection methods such as PCR, using sufficient volumes of tissues as a matrix, might be optimal to obtain an indication of the proportion of animals presenting a risk for human infection in these species.

The aim of this study was to obtain age-dependent prevalence estimates in selected animal species. To this end, data on both indirect and direct detection methods were extracted from the literature, and a meta-analysis was carried out using a hierarchical Bayesian age-dependent model including relevant covariates.

## 2. Materials and Methods

### 2.1. Data

The literature review was conducted, and the results were reported following the PRISMA guidelines [23]. A structured literature search was carried out on 16 June 2020, covering the literature published up to this date using the Embase literature database. A search string (see Appendix A) was developed based on previous work [22] using Emtree terms. The search was designed to cover farm animal and wildlife species intended for human consumption, as well as feline species, sampled within Europe. Forty-one countries, including the 27 European Union member states, were included in the search strategy. The selection of included animal species was based on the FoodEx2 database [24] by including all animal species with consumption data from national food consumption surveys from the years 1997–2018.

The eligibility of retrieved publications was assessed by a team of 20 scientists from 13 countries across Europe using Cadima [25], an open-access online tool for conducting systematic reviews. The screening of publications was based on a set of predefined criteria. Only peer-reviewed articles featuring original data on the detection of *T. gondii* (e.g., prevalence studies, epidemiological surveys) using both direct and indirect methods in selected host species in Europe, published in English, with at least a part of the data collected from the year 2000 onwards, were considered eligible. Experimental infection studies, case-control studies, literature reviews, meta-analyses, books, conference proceedings, grey literature, and publications with incomplete data necessary for the modelling (unreported sampling period, animal species, number of total and positive animals, or country), or cases where the full-text could not be obtained, were excluded. Each publication was screened by two randomly chosen scientists from the group, first at the level of the title and abstract, and after a consensus on inclusion between the two scientists was reached, full-text screening was performed on the remaining publications.

Publications that met the inclusion criteria after full-text screening proceeded to data extraction, which was conducted by a smaller group of nine scientists. For each publication, the data were extracted by one of the nine scientists, and inputs were checked by another scientist. Inconsistencies and disagreements were discussed until an agreement was reached. A data extraction template file (see Appendix A) was created in a spreadsheet (Microsoft Excel) to record the required data. For each study, data describing the study design, sampling, and testing methods were collected as follows: Country or the region covered in the epidemiological screening, animal species, total number of farms or herds sampled, the total number of animals sampled, animal age or age group estimates, first and last years of the sampling period, and the sample type used. Data on sample testing encompassed the type of diagnostic assay, the commercial test name (when applicable), and cut-off values. Extracted data were harmonized and categorized for modelling. Firstly, animal species were sorted into thirteen categories (Table 1) based on their common physiological traits and phenotype.

As grouping on a coarser level is more suitable for hierarchical modelling, the countries present in the dataset were assigned to one of five European regions—Western, Northern, Eastern, Southeastern, and Southwestern (Figure 1, Appendix A), as described previously [26].

The detection methods described in the publications were classified as direct or indirect, depending on whether the parasite (or its DNA) or the antibody response is detected. The matrices tested were grouped into three categories—“blood”, “meat juice”, and “other”. The blood category comprised mostly serum samples but also included whole blood and plasma samples. The meat juice category consisted of liquid samples obtained by freezing and thawing muscle tissues. The “other” category incorporated mainly matrices used for direct detection (e.g., organ and muscle tissues, milk, and, in the case of felids, feces) but also pleural fluids and cardiac fluids used for indirect detection (see Appendix A).

Some publications presented more than one set of results on *T. gondii* detection for a group of animals (e.g., results based on multiple indirect methods, different matrices, or with multiple cut-off values), resulting in several rows of data for the same population in the spreadsheet. In order to avoid counting the same animal population multiple times, a population identifier was introduced. A population was defined as a set of animals of the same species, age group, and from the same study. For each population identifier, a weighting was derived as one over the number of entries. This weighting was used in the model to scale the influence of the entry. Weighting is similarly introduced in case two direct methods are used; however, the number of entries for indirect and direct are considered separately.

Indoor animals were defined as those kept strictly in an enclosed environment (e.g., pigs kept under controlled housing) and outdoor-held animals as animals commonly kept in the open, with potential contact with the outside environment and free-roaming felids. Indoor and outdoor access were based on details from the publication or the typical holding and living conditions (e.g., strictly outdoor access for wildlife). For buffalo, cattle, equids, sheep, and game species (wild boar, wild birds, and wild ruminants), outdoor access was assumed by default. For publications that included data on animals with and without outdoor access, these data were extracted on separate rows and given a different population identifier.

Since the ages of individual animals at the time of sampling were not given, we define an uncertainty distribution based on estimates of the minimum, maximum, and most probable age at sampling. The most probable age at sampling often coincides with the slaughter age. Age ranges, with the minimum, maximum, and most probable age at sampling, were extracted from publications whenever available. In case only the age range was indicated in the publication, for livestock, the arithmetic mean of the provided age range was used as a default value for the most probable age at sampling. For wildlife, in cases where age information was missing in the publication, an estimate for the minimum, maximum, and most probable age of sampling was applied based on the maximum recorded age for the animal species while taking into account the most common hunting age. Details can be found in the Appendix A).

### 2.2. Analysis of Direct Detection Data

The amount of data from direct tests was insufficient to parameterize the full Bayesian model. Instead, the prevalence was calculated per species by dividing the number of positive animals by the total number of tested animals and multiplied by the population weighting factor (for details see Appendix A). Confidence intervals were based on the binomial distribution.

### 2.3. Analysis of Indirect Detection Data

#### 2.3.1. Age-Structured Model

Indirect tests do not provide a good indication of the presence of infective *T. gondii* in cattle, buffalo, and equids, therefore these species were excluded from the age-dependent seroprevalence modelling. Two different compartmental infection models were considered to fit the age-dependent prevalence data based on indirect tests:

(1) The susceptible-infected (SI) model where animals move from susceptible (i.e., seronegative) to infected (i.e., seropositive), and (2) the susceptible-infected-susceptible (SIS) model, where we also allow for a reversion to susceptible (i.e., loss of detectable antibody response).

In both of these models, animals are born susceptible and can move into the ‘infected’ compartment based on a constant force of infection λ (incidence rate, measured in new infections per year per animal). The SIS model also allows for reversion to seronegative with a rate of *γ*.

The variables S and I are fractions of the total population, dependent on age a, adding up to one for each age. For the SIS model, the equation for the prevalence is
(1)$$I_{SIS}\left(a\right)=\frac{\lambda }{\lambda +\gamma }\left[1-\exp\left(1-\left(\gamma +\lambda \right)a\right)\right]$$

The SI model was recovered by setting γ=0.

#### 2.3.2. Bayesian Hierarchical Model

A Bayesian model was built to estimate the prevalence in different animal populations and regions to quantify uncertainty. Moreover, a hierarchical model was built to be able to estimate the prevalence in countries with no data by “borrowing” estimates through partial pooling, granting us the possibility to overcome the data gaps. The Bayesian model was built based on the variables as shown in Table 2 (for model details, see Appendix A).

The basis of the Bayesian model was to describe the number of positive animals as a realization of a random process where each individual of the population of size ntot[i] had a probability I(a) to be found infected, resulting in npos[i] positives:(2)npos[i]~Binomial(ntot[i],I(a[i]))

The function I(a) is from either the SI or SIS model described above. The age-distribution a[pop[i]] of each population pop[i] was a beta distribution, scaled to lie between agemin[pop[i]] and agemax[pop[i]],
(3)a∗[pop[i]]~Beta(α[i],β[i]),
(4)$$a\left[pop\left[i\right]\right]=\frac{a*\left[pop\left[i\right]\right]-agemin\left[pop\left[i\right]\right]}{agemax\left[pop\left[i\right]\right]-agemin\left[pop\left[i\right]\right]}.$$

That way, the age-distribution set for each population will be updated to facilitate the model fit in the posterior age distribution. As a prior the mean was set using μ[i]=α[i]/(α[i]+β[i]), giving the best estimate of the age age[i], and the precision ϕ[i]=α[i]+β[i] to a low value of 10.

Differences between species, outdoor access, sample type, and regions were taken into account using a hierarchical model. These differences were modelled as linear contributions to the logarithmic force of infection:(5)$$\log\left(\lambda_{i}\right)=\lambda_{0}+\lambda_{species\left[i\right]}+\lambda_{outdoor\left[i\right]}+\lambda_{sampetype\left[i\right]}+\lambda_{region\left[i\right]}$$
$$\lambda_{species\left[i\right]}\sim N\left(0,\sigma_{\lambda_{species}}\right),\;\lambda_{outdoor\left[i\right]}\sim N\left(0,\sigma_{\lambda_{housing}}\right),$$
$$\lambda_{sampletype\left[i\right]}\sim N\left(0,\sigma_{\lambda_{sampletype}}\right),\;\lambda_{region\left[i\right]}\sim N\left(0,\sigma_{\lambda_{region}}\right).$$

All hierarchically modelled forces of infection are soft-centered at zero to render the model identifiable. The parameter $\lambda_{0}$ can then be regarded as a baseline force of infection, which was given a vague prior $\lambda_{0}\sim N\left(1,\;2\right)$. All standard deviations are supplied with vague priors. We assume that the loss of detectable antibodies is a physiological process, which only depends on species, and modelled *γ* analogously to *λ*,
(6)$$\log\left(\gamma_{i}\right)=\gamma_{0}+\gamma_{species\left[i\right]},\;\gamma_{species\left[i\right]}\sim N\left(0,\sigma_{\gamma_{species}}\right)$$

To prevent multiple counting of populations and resulting artificial inflation of precision, we weighted contributions to the likelihood using the weighting factor defined before.

Model fitting was performed using Stan [27] interfaced from R v4.1.3. [28]. Trace plots of the Markov chains were visually assessed to confirm the convergence of the model (see Appendix A).

## 3. Results

### 3.1. Data Collection

A total of 1985 publications were retrieved, with more than 50 animal species included (Table 1). Twenty-four articles were excluded as duplicates, 1599 publications did not meet the inclusion criteria during the title and abstract screening, and a further 86 were excluded during the full-text screening (see Appendix A). Following the screening process, 276 publications were considered eligible for data extraction, out of which 226 publications with a complete set of data were included. Relevant data on at least one of the animal species of interest were recovered from 29 out of the 41 countries included in the search string. The number of articles on each of the individual species, including the number of animals and the type of detection method used, is summarized in the Appendix A). Direct testing methods included various molecular methods (PCR, quantitative PCR, nested PCR, and magnetic capture PCR) and other *T. gondii* parasite detection methods (bioassay, direct immunofluorescence test, direct Western Blot, flotation, and sedimentation followed by microscopy). The indirect methods used to detect specific anti-*T. gondii* IgG antibodies included the modified agglutination test, the direct agglutination test, the latex agglutination test, ELISA, the indirect fluorescence test, the Sabin–Feldman dye test, and the indirect Western Blot.

### 3.2. Animal Prevalence Results of Direct Methods

The highest prevalence based on direct detection data was 68.4% (95% CI: 54.8–80.1%) observed in the brains of lagomorphs in the Western region (other tissues, the United Kingdom) (Figure 2, Appendix A). The lowest *T. gondii* prevalence was observed in the hearts of equids from the Eastern region (*n* = 82, other tissues, Romania) at 0.0% (95% CI: 0.0–0.4%) (Figure 2, Appendix A). A low prevalence, equal to 0.0% when rounded up to a single decimal space, was also recorded in buffaloes from the Eastern region (*n* = 74, 95% CI: 0.0–4.9%), lagomorphs from the Southeastern region (*n* = 52, 95% CI: 0.0–6.9%), pigs from the Southwestern region (*n* = 44, 95% CI: 0.0–8.0%), and wild birds from the Southwestern region (*n* = 5, 95% CI: 0.0–52.2%). A consistently low *T. gondii* prevalence was observed in fecal samples of felids, especially in the Northern region at 0.2% (*n* = 598, 95% CI: 0.0–0.9%) and the Western region at 0.2% (*n* = 104309, 95% CI: 0.1–0.2%). Average weighted prevalence estimates for included animal species reported separately for different sample matrices can be found in Appendix A (Appendix A). No eligible direct detection data were available for ducks and geese and poultry.

### 3.3. Age-Dependent Animal Seroprevalence Results Using the Bayesian Hierarchical Model

Trace plots and comparison of between and within chain estimates indicated good convergence of the model (see Appendix A).

The effects of region, species, sample type, and outdoor access on the exponentiated force of infection and reversion rate are visualized in Figure 3. The baseline force of infection ($\exp\left(\lambda_{0}\right)=$ 1.92, 95% CI: 1.15–2.93) and the reversion rate ($\exp\left(\lambda_{0}\right)=$ 0.25, 95% CI: 0.07, 0.53) express the average with regards to all other components. The effect of these posteriors is multiplicative, with the mean value set to 1. The variation over regions was the lowest ($\sigma_{\lambda_{region}}=$ 0.36, 95% CI: 0.16–0.84), with the difference in the force of infection over the sample type ($\sigma_{\lambda_{sampletype}}=$ 0.51, 95% CI: 0.16–1.41) and outdoor access ($\sigma_{\lambda_{outdoor}}=$ 0.65, 95% CI: 0.23–1.66) being slightly higher. The largest variation was found for an interspecies force of infection ($\sigma_{\lambda_{species}}=$ 0.85, 95% CI: 0.54–1.40) and reversion rates ($\sigma_{\gamma_{species}}=$ 1.92, 95% CI: 1.15–2.93) (Figure 1). Given the baseline force of infection of $\exp\left(\lambda_{0}\right)=1.92$ and the reversion rate $\exp\left(\gamma_{0}\right)=0.25$, these are all considerable contributions, as can also be visually assessed from Figure 3.

For the force of infection or reversion, one over the posterior coefficient can be interpreted as an average waiting time in years until the event. For the force of infection, this yields a waiting time of 11.92≈ 0.5 years until *T. gondii* infection and a waiting time of 10.25≈ 4 years on average to become susceptible (seronegative) again.

The model results show clear differences in the force of infection between the animal species (Figure 3). In four of the ten groups (pigs, poultry, lagomorphs, and wild birds), the force of infection was below the overall average (lowest in lagomorphs $\exp\left(\lambda_{species}\right)=0.24$, 4 years waiting time). Six animal groups (ducks and geese, felids, goats, sheep, wild boars, and wild ruminants) had a higher force of infection than average (highest for felids $\exp\left(\lambda_{species}\right)=$ 3.53 corresponding to a waiting interval of just over three months). In some animal species, especially those with a higher force of infection, a high uncertainty was observed due to a lack of data. Note that these estimates do not include the effects of the other parameters and do not take into account, for example, if species are held indoors or outdoors. To incorporate this aspect, the result must be multiplied by $\exp\left(\lambda_{outdoor}\right)$ or $\exp\left(\lambda_{indoor}\right).$

Regarding reversion, four animal groups (chickens and hens, lagomorphs, pigs, and wild ruminants) scored a reversion rate lower than the European average of approximately four years to become susceptible again. The shortest reversion rate on average was calculated for pigs at six months ($\exp\left(\gamma_{species}\right)=$ 0.1). Six animal groups (ducks and geese, felids, goats, sheep, wild boars, and wild ruminants) had a reversion rate above the calculated average, with the longest interval of approximately seven years, observed in felids ($\exp\left(\gamma_{species}\right)=$ 3.7) (Figure 3).

The lowest overall force of infection across the species’ spectrum within the European regions was observed in the Western region ($\exp\left(\lambda_{region}\right)=$ 0.3), followed closely by the Northern region ($\exp\left(\lambda_{region}\right)=$ 0.6). The Southwestern ($\exp\left(\lambda_{region}\right)=$ 1.3) and Southeastern regions ($\exp\left(\lambda_{region}\right)=$ 1.4) showed a similar force of infection, slightly above the European average. The highest force of infection was observed in the Eastern region ($\exp\left(\lambda_{region}\right)=$ 3.5) (Figure 3). It is estimated that the time until encountering *T. gondii* infection in animals from the Eastern region averages approximately three months, which would be more than eight times shorter than in the Western region (approximately 3 years).

The force of infection was almost five times higher in outdoor-held animals ($\exp\left(\lambda_{outdoor}\right)=$ 2.1) than in those kept strictly indoors ($\exp\left(\lambda_{indoor}\right)=$ 0.4) (Figure 3). The time until *T. gondii* infection was, on average, 6 months for outdoor-kept animals and 2.6 years for indoor-kept animals. Animals with unknown holding backgrounds are similar to those with outdoor access ($\exp\left(\lambda_{outdoor}\right)=$ 2.1).

Model results indicate that meat juice samples were almost five times as likely to be positive for *T. gondii* antibodies than blood samples (Figure 3). The force of infection of other matrices ($\exp\left(\lambda_{sampletype}\right)=$ 0.6) yielded results closer to the blood sample type ($\exp\left(\lambda_{sampletype}\right)=$ 0.7) than the meat juice sample type ($\exp\left(\lambda_{sampletype}\right)=$ 3.4).

An example of the age-dependent seroprevalence curves for pigs is shown in Figure 4. Similar curves are available for all species (see Appendix A) and are used to estimate the seroprevalences at the age most relevant for human infection, split by outdoor access of the animal as shown in Table 3. The highest seroprevalence was estimated for sheep with outdoor access in the Eastern region at 78.5% (95% CI: 77.0–79.8%; 3.7 years of age) (Table 3). In contrast to the result of direct methods, the lowest seroprevalence was estimated for indoor-kept lagomorphs in the Eastern region at 2.0% (95% CI: 1.7–2.4%; 0.2 years of age). *Toxoplasma gondii* seroprevalence estimates were the highest in the Eastern region, followed by the Southeastern and Southwestern and the lowest in the Northern and Western regions.

## 4. Discussion

An extensive literature review on the prevalence of *T. gondii* infection in animal species that can be relevant as sources of human infection was carried out. Although the prevalence of *T. gondii* in animals does not give a direct indication of the risk of human infection, the prevalence in combination with exposure data is important to estimate the relative contribution of different sources of *T. gondii* human infection by quantitative risk assessment. Performing the literature search within the Embase database allowed us to utilize its advanced guided mapping of keywords to Emtree, conveniently covering a broad range of animals and countries in the literature search. In the present study, the animal prevalence was modelled using a Bayesian approach, which is increasingly being used in epidemiological studies because of the ability to quantify uncertainty. Moreover, hierarchical modelling allows handling cases with little or no data by “borrowing” estimates through partial pooling, granting us the possibility to overcome the data gaps [29].

In this review, the modelled outcomes based on serological data show the lowest overall seroprevalence of *T. gondii* in indoor-kept lagomorphs at 4.8% (95% CI: 1.8–7.5) and the highest in outdoor-kept sheep at 63.3% (95% CI: 53.0–79.3). Previous attempts at estimating *T. gondii* seroprevalence in animals using meta-analyses were mostly bound to a single country outside of Europe [30,31] or focused on a limited number of animal species [32,33,34]. The only recently published meta-analysis overlooking the European livestock and poultry seroprevalence data within a comparable time period provides only a single combined seroprevalence (43.5%, 95% CI: 32.1–55.6%) for all included species [35]. It is not justified or possible to validate the outcomes of the present model by comparison to original seroprevalence studies in animals in Europe because these studies were the input for our model.

Data based on direct testing methods in Europe is relatively scarce and insufficient for inclusion in the Bayesian hierarchical model. Therefore, the results of studies using direct detection methods are summarized separately with no age-modelling applied to them (Figure 2, Appendix A). A comparable study performed on a global scale and using data from direct methods showed the lowest pooled prevalence in cattle, followed by pigs, and the highest in sheep for Europe as a whole [36]. Similarly, a low prevalence of *T. gondii* in cattle was determined in the current study; however, the prevalence in pigs and sheep varied greatly between the regions and matrices included (see Appendix A).

Seroprevalence results may be a more representative indicator than direct methods for *T. gondii* infection as tissue cysts are not homogenously distributed within infected animals and may go undetected [22]. Nonetheless, sheep, which have the highest prevalence by direct detection according to Belluco et al., 2016 [36], also have the highest seroprevalence estimates modelled for outdoor-kept sheep in Europe.

The present model is methodologically the closest to the published model of Deng et al., 2018 [31] who introduced two of the covariates presented in the current model—animal species and geographical region. Additionally, the current model incorporates the effect of the sample type used and the outdoor access of the animal. The regional variation, introduced previously by Deng et al., 2018 [31], was modified to fit the needs of the present study. Even with the variation over regions being the lowest among all the factors included ($\sigma_{\lambda_{\mathrm{region}}}=$ 0.36, 95% CI: 0.16–0.84), the differences between regions were considerable, justifying the reporting of the seroprevalences per region. The Bayesian hierarchical modelling made it possible to fill the data gaps present for some combinations of species and regions. Future model extensions for incorporating regional variation could involve the use of spatially explicit modelling as presented by Gotteland et al., 2014 [37], or the use of automated clustering to be performed directly by the model [38].

For the majority of animal species, the seroprevalence of *T. gondii* is known to increase with age [14,20,39]. Therefore, the most important addition to the current model is the introduction of a dynamic transmission (SIS) model to allow the inclusion of data reported for heterogeneous age ranges in one model. This approach, using a compartmental infection model, was applied previously in age-dependent modelling for estimating *T. gondii* prevalence in animals [40,41]. It should be noted that the “I” in our epidemiological model stands for “Infected” or seropositive rather than “Infectious”, as the majority of *T. gondii* infections in animals occur due to an external source of infection (e.g., oocysts or infected prey) instead of via transmission between animals within one animal population [13,14]. The dogma of a lifelong persistence of *T. gondii* antibodies [42], where susceptible animals move to the infected compartment and the seroprevalence always reaches 100% if animals live long enough, corresponds to an SI epidemiological approach [43,44]. However, the SIS approach with a reversion rate was better able to fit the plateau observed in the seroprevalence data at high age. The reversion rate did not revert to zero in any of the species and was lowest in pigs ($\exp\left(\gamma_{species}\right)=$ 0.1). Alternatively, an SI model could also reach a plateau of less than 100% when a subpopulation of animals is not exposed or does not develop a detectable antibody response. To settle these hypotheses, animals would have to be sampled and monitored multiple times in life.

Another feature of the model is the estimation of the effect of outdoor access on *T. gondii* seroprevalence. Our results show an almost five times higher force of infection in outdoor-kept animals compared to those kept strictly indoors. These findings agree with the outdoor access of animals being one of the most commonly identified risk factors for acquiring *T. gondii* infection [14,15,16]. This outcome can be explained by a higher infectious pressure caused mostly by environmental contamination with oocysts and the presence of potentially infected rodents and birds that could be preyed on [13,45,46]. As the presence of cats on farms is also a well-known risk factor for *T. gondii* infection in animals [13,14], ideally, this should be taken into consideration in the modelling, but this information is only rarely reported. For the final results, with prevalence by region at the most relevant age, it was, unfortunately, not possible to combine the prevalence into a weighted prevalence with regards to indoor/outdoor access due to insufficient data on the proportions of indoor to outdoor-kept animals in the different regions.

Variability in the force of infection over sample type ($\sigma_{\lambda_{\mathrm{sampletype}}}=$ 0.51, 95% CI: 0.16–1.41) was observed. Surprisingly, the results show that the serological testing of meat juice samples was almost five times more likely to result in *T. gondii* antibody detection than when using blood or other matrices (milk, pleural fluid, and cardiac fluid) (Figure 3), suggesting a higher sensitivity of meat juice matrix. Thirteen studies provided data on *T. gondii* detection using meat/muscle juice; however, sera of the same animals were never tested in parallel. Even though meat juice can be considered an alternative matrix to serum for antibody detection [47], performing indirect tests using serum samples appears to provide more reliable results than other matrices such as meat juice, where the concentration of the specific antibodies is less homogenous and depends on the muscle the meat juice has been extracted from [48]. Due to the lack of parallel testing of the two matrices, we cannot exclude the possibility of selection bias (i.e., meat juice has been used more often in populations with a higher risk of *T. gondii* infections) and consider blood serum as a matrix of choice for the specific anti-*T. gondii* antibody detection, despite the modelled outcome. More data from a parallel indirect screening of samples from the same animal is needed to establish diagnostic test sensitivity and specificity of indirect tests when using different matrices. Adding these data as prior information could be a useful addition to the current model.

In the study by Deng et al., 2018 [31], separate literature reviews were carried out for the different serological tests used in order to take into account test characteristics in the seroprevalence modelling and provide corrected true prevalence estimates. A similar approach was considered not feasible for animal studies, as there is more diversity in serological tests applied and a lack of harmonized studies on performance characteristics in comparison to the assays used in human diagnostic laboratories. Moreover, in order to include as many data as possible, we did not exclude studies based on the test or cut-off value used and included the results as presented by the authors. The uncertainty in the Bayesian framework consists of model uncertainty and parameter uncertainty. Model uncertainty refers to the fact that the exact processes governing reality are unknown. Most pertinent in the current case is the choice between an SI model and an SIS model in light of the observed plateau in seroprevalence by age, as explained above. To deal with parameter uncertainty, the parameters of the model are supplied with (prior) uncertainty distributions. After running the model with the data, output (posterior) uncertainty distributions are obtained. The uncertainty present due to the lack of data for the covariates is reflected in the posterior probability intervals from the Bayesian hierarchical model. For the regions or species with insufficient data, the Bayesian hierarchical model allowed us to obtain seroprevalence estimates based on the data from the remaining regions or species but with larger uncertainty than those with data. Furthermore, the uncertainty was affected by the coarse granularity and missing metadata of reported data. Our suggested way of dealing with this is by following a single data reporting template, which includes all the necessary (meta)data. We would like to stress the importance of publishing a full set of raw data and adhering to a set of fixed guidelines for reporting. Preferably, data should also be published on a per-animal basis, for example in an online supplement. A simple spreadsheet template for epidemiological data reporting that could act as such a guideline was developed (see Appendix A). Individual rows should be used to report data on individual animals rather than populations. The template itself is divided into three parts (sampling, processing, and reporting) and facilitates data reporting and the future use of the data for reviews and meta-analyses, thus increasing the visibility and reach of the data and corresponding articles. Instead of only mentioning results based on a fixed cut-off value, it is advised to provide OD values or titers so a custom cut-off can be applied by the user. Ultimately, the creation of a single publicly accessible online data repository incorporating homogenously structured data from original research studies would be an invaluable resource for future studies.

The current study describes a literature review and meta-analysis using a complex Bayesian hierarchical model including a novel dynamic transmission model, based on data obtained from an extensive meta-analysis. Age as well as all other covariates, such as indoor/outdoor status and region, can be extrapolated to obtain prevalence estimates for the actual composition of the animal population (with regards to animal species, outdoor access, etc.) in any European country despite data gaps. Prevalence estimates presented in this study, as well as those obtained from the model by users, could be used as a valuable tool for various purposes such as calculating adequate sample sizes for serological screening based on the seroprevalence in the region, a comparison tool, and could even provide a powerful instrument for the policymakers, e.g., to evaluate consequences of slaughtering age on the risk of *T. gondii* infection. Moreover, the modelled seroprevalence estimates, together with consumption data of the meat products of these animals, are a starting point for future risk assessments and are being implemented as such in a multi-country quantitative microbial risk assessment for *T. gondii* source attribution in humans.

## Figures and Tables

**Figure 1 pathogens-12-00097-f001:**
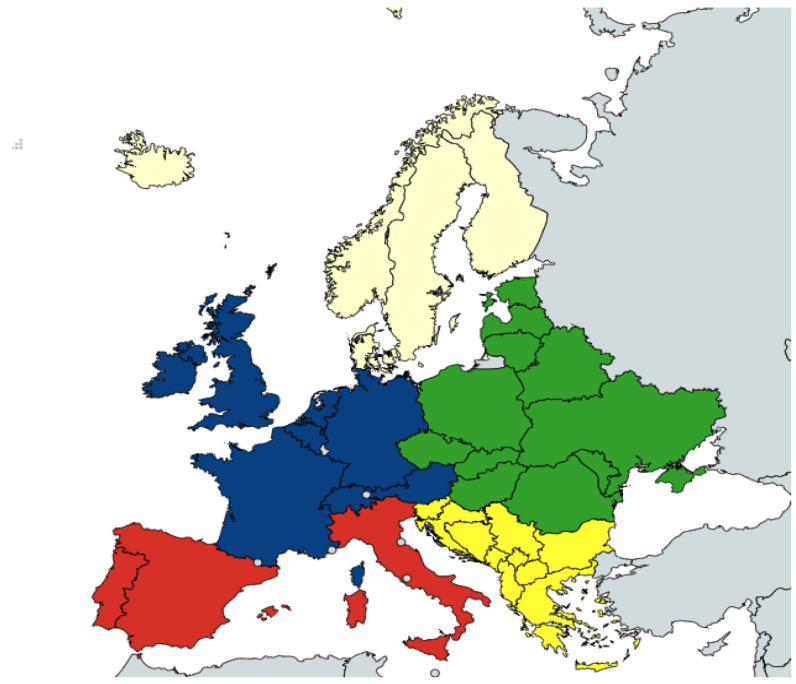
Categorization of Europe into regions. European countries are categorized into five European regions: Western (blue), Northern (white), Eastern (green), Southeastern (yellow), and Southwestern (red). Generated from: https://www.mapchart.net/index.html, accessed on 18 August 2022.

**Figure 2 pathogens-12-00097-f002:**
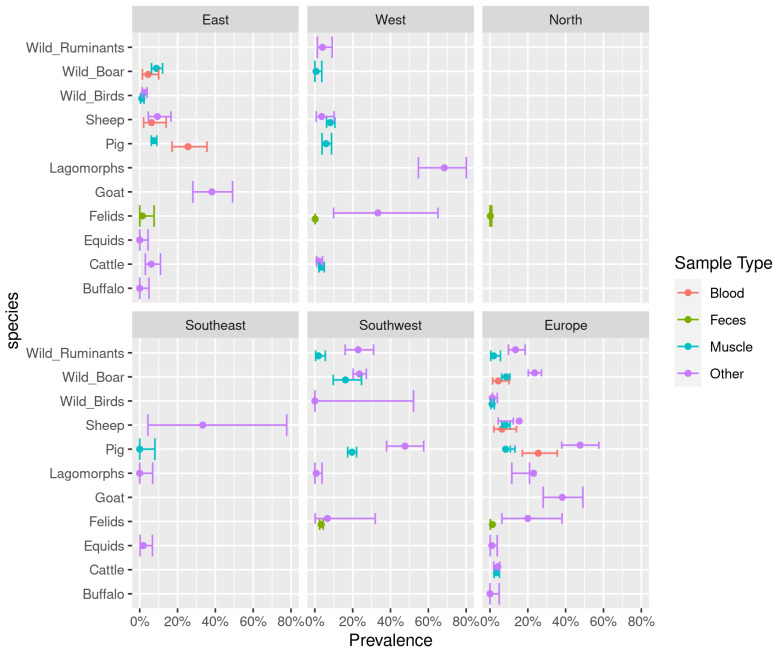
Direct detection estimates. Average weighted prevalence estimates with 95% CI per region based on direct detection and separated by sample types of blood, feces, muscle, and other types (muscle and organ tissues, milk, pleural fluids, cardiac fluids, and, in the case of felids, feces). Only animal species that had data available from at least one of the five regions are included.

**Figure 3 pathogens-12-00097-f003:**
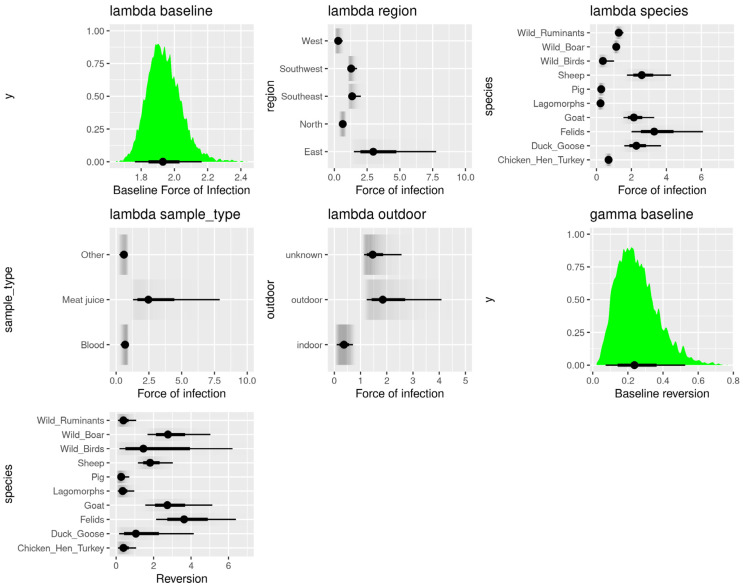
Posterior probabilities of the model parameters. Values for the force of infection in regions, in species, by sample type and outdoor access are all exponentiated, which means that a value of one indicates the absence of an effect on the baseline probability (see Appendix A). To reconstruct the total force of infection, the exponentiated contributions must be multiplied. Gray area represents the uncertainty distribution. In all panels, the thin and thick black lines indicate 95% and 50% Bayesian credible intervals, respectively, with dots indicating the mean of exponentiated forces of infection dependent on region, species, sample type (other = pleural or cardiac fluids), and holding status as multiplicative corrections to the baseline (see Equation (6)).

**Figure 4 pathogens-12-00097-f004:**
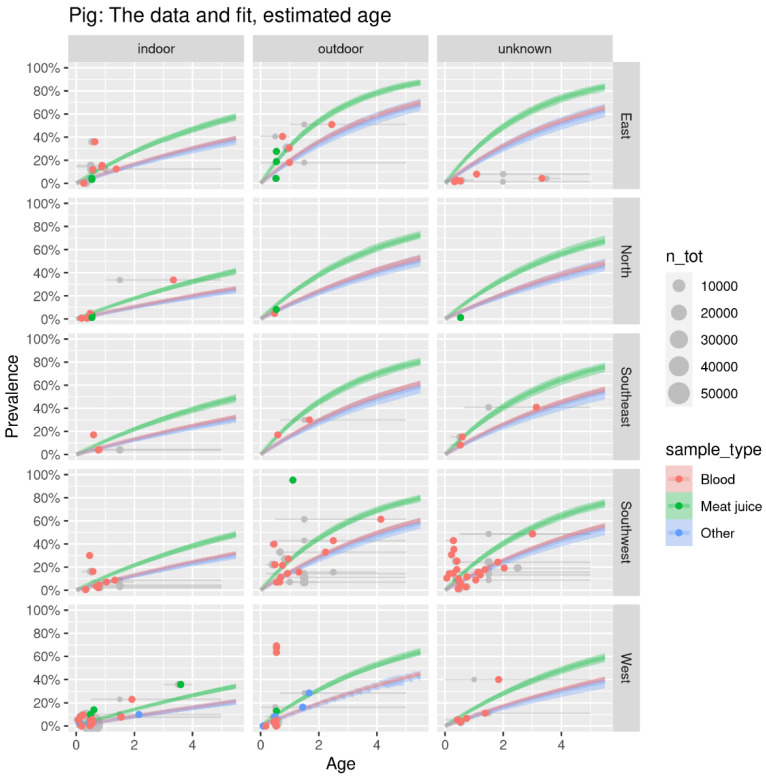
SIS model fit for age-dependent seroprevalence of *T. gondii* in pigs by outdoor access in five regions of Europe. The lines indicate the fitted seroprevalence by age, based on detection in the blood (red), muscle fluid (green), or other matrices (blue). The grey dots represent seroprevalence data points at the best estimate of age in the data for the studied populations. The size of the dots indicates number of animals, in categories from 10,000 to 50,000. The dots are shifted horizontally along the grey line extending from the minimum to maximum possible age, to the best fitting age (red dots) (see Appendix A).

**Table 1 pathogens-12-00097-t001:** Animal categories and applied testing methods. Animal categories with associated animal species (animal species as defined by ITIS, Integrated Taxonomic Information System) and the types of test methods considered relevant to obtain prevalence estimates (direct methods demonstrate the presence of the parasite, and indirect methods demonstrate the presence of antibodies).

Group	Animal Species Included	Testing Methods
Buffalo	*Bubalus bubalis*	Direct
Cattle	*Bos taurus*	Direct
Duck/Goose	*Anas platyrhynchos, Anser anser, Anser cygnoides*	Indirect
Equids	*Equus caballus, Equus asinus* and their cross-breeds	Direct
Felids	*Felis catus, Felis silvestris, Lynx lynx, Lynx pardinus*	Direct and Indirect
Goat	*Capra hircus*	Direct and Indirect
Lagomorphs	*Oryctolagus cuniculus, Lepus europaeus, Lepus granatensis, Lepus timidus*	Direct and Indirect
Poultry	*Galus galus, Meleagris gallopavo*	Indirect
Pig	*Sus scrofa*	Direct and Indirect
Sheep	*Ovis aries*	Direct and Indirect
Wild birds	*Anas crecca, Aythya ferina, Anas penelope, Anas strepera, Anas platyrhynchos* (feral), *Anas acuta, Anas clypeata, Phasianus colchicus, Columbidae* (family), *Anas platyrhynchos* (feral)	Direct and Indirect
Wild boar	*Sus scrofa* (feral)	Direct and Indirect
Wild ruminants	*Rupicapra rupicapra, Cervidae* (family), *Dama dama, Alces alces, Ovis aries musimon, Ovis gmelini musimon, Ovis musimon, Ovis orientalis musimon, Ovis aries, Ovis ammon, Cervus elaphus, Rangifer tarandus, Rangifer tarandus platyrhynchus, Capreolus capreolus, Cervus nippon, Capra pyrenaica hispanica, Capra pyrenaica victoriae, Capra pyrenaica, Odocoileus virginianus*	Direct and Indirect

**Table 2 pathogens-12-00097-t002:** Data used in the Bayesian hierarchical model. Variables included in the model with corresponding values.

Variable	Values
species[*i*]	Buffalo; Felids; Cattle; Duck, Goose; Goat; Equids; Pig; Chicken, Hen, Turkey ^a^; Lagomorphs; Sheep; Wild Birds; Wild Boar; Wild Ruminants
region[*i*]	East, North, Southeast, Southwest, West
pop[*i*]	A unique identifier for a population
test[*i*]	Direct, Indirect
outdoors[*i*]	Outdoor, Indoor, Unknown
sample_type[*i*]	Blood, Meat juice, Other ^b^
ntot[*i*]	Total number of animals tested
npos[*i*]	Total number of animals found positive
age[*i*]	Best estimate of average age range
agemin[*i*]	Lower bound of the age range
agemax[*i*]	Upper threshold of the age range
agemean[i]	The most probable age at sampling

^a^ referred to as poultry in the following, ^b^ organ- and muscle tissues, milk, pleural fluids, cardiac fluids and feces.

**Table 3 pathogens-12-00097-t003:** Modelled regional seroprevalence estimates. Regional seroprevalence estimates (sample type = “blood”) were modelled for selected animal species groups at their most probable age of sampling, reported separately for indoor and outdoor holding conditions. Numbers in square brackets are the ages most relevant for human infection. The asterisk indicates the age of the animal species that have been estimated from the demographics of the other countries within the region.

Species	Holding Conditions	East	North	Southeast	Southwest	West	Europe
Chicken_Hen_Turkey	indoor	10.3% (9.2%, 11.6%) [0.6]	10.4% (9.2%, 11.8%) [0.9 *]	13.0% (11.5%, 14.7%) [0.9 *]	1.6% (1.4%, 1.8%) [0.1]	3.4% (3.1%, 3.9%) [0.4]	7.8% (1.5%, 14.0%)
	outdoor	23.8% (21.5%, 26.3%) [0.6]	23.5% (21.1%, 26.2%) [0.9 *]	28.8% (25.9%, 32.0%) [0.9 *]	6.3% (5.6%, 7.0%) [0.2]	27.4% (25.1%, 30.0%) [1.4]	22.0% (5.9%, 30.7%)
Duck_Goose	indoor	10.2% (8.8%, 11.8%) [0.2]	11.4% (9.7%, 13.4%) [0.4 *]	14.3% (12.1%, 16.7%) [0.4 *]	14.0% (11.9%, 16.2%) [0.4 *]	5.1% (4.4%, 5.9%) [0.2]	11.0% (4.7%, 15.9%)
	outdoor	25.6% (22.3%, 29.3%) [0.2]	25.5% (21.9%, 29.6%) [0.4 *]	31.2% (26.9%, 35.9%) [0.4 *]	30.7% (26.7%, 35.1%) [0.4 *]	31.0% (26.2%, 35.7%) [0.6]	28.8% (22.8%, 35.1%)
Felids	indoor	48.0% (45.2%, 50.9%) [7.3]	27.2% (24.7%, 29.6%) [4.6 *]	32.9% (30.1%, 35.7%) [4.6 *]	24.7% (22.3%, 27.1%) [3.0]	26.5% (24.8%, 28.2%) [6.2]	31.9% (23.3%, 49.6%)
	outdoor	73.8% (70.9%, 76.6%) [7.5]	41.2% (37.6%, 44.7%) [2.9]	49.9% (46.1%, 53.6%) [3.0]	54.0% (50.9%, 56.9%) [3.7]	51.5% (49.3%, 53.8%) [6.6]	54.1% (39.1%, 75.5%)
Goat	indoor	26.5% (24.1%, 28.9%) [2.8 *]	17.4% (15.7%, 19.1%) [2.8 *]	21.6% (19.7%, 23.5%) [2.8 *]	21.2% (19.2%, 23.1%) [2.8 *]	16.0% (14.5%, 17.5%) [3.3]	20.5% (15.1%, 27.9%)
	outdoor	47.5% (44.2%, 50.6%) [2.4]	46.0% (42.9%, 49.1%) [4.0]	44.0% (41.1%, 46.6%) [2.8]	48.2% (45.2%, 51.0%) [3.3]	30.6% (28.2%, 32.9%) [2.8 *]	43.3% (29.2%, 50.2%)
Lagomorphs	indoor	2.0% (1.7%, 2.4%) [0.2]	5.4% (4.6%, 6.3%) [1.1 *]	6.8% (5.7%, 8.0%) [1.1 *]	5.4% (4.6%, 6.3%) [0.9]	4.3% (3.7%, 5.0%) [1.1 *]	4.8% (1.8%, 7.5%)
	outdoor	18.8% (16.3%, 21.6%) [1.1]	16.8% (14.4%, 19.4%) [1.5]	20.8% (17.9%, 24.0%) [1.5]	16.0% (13.9%, 18.3%) [1.1]	12.1% (10.5%, 13.9%) [1.3]	16.9% (11.1%, 22.7%)
Pig	indoor	5.2% (4.9%, 5.4%) [0.6]	3.6% (3.3%, 3.8%) [0.6]	8.3% (7.7%, 8.9%) [1.2]	6.6% (6.3%, 6.9%) [1.0]	2.4% (2.4%, 2.5%) [0.5]	5.2% (2.4%, 8.7%)
	outdoor	22.0% (21.1%, 23.0%) [1.1]	6.5% (6.0%, 7.0%) [0.5]	16.9% (15.8%, 18.1%) [1.0]	18.5% (17.7%, 19.3%) [1.2]	5.5% (5.3%, 5.7%) [0.5]	13.9% (5.4%, 22.6%)
Sheep	indoor	43.9% (41.8%, 46.1%) [3.2 *]	30.3% (28.4%, 32.3%) [3.2 *]	36.7% (34.5%, 38.9%) [3.2 *]	36.1% (34.5%, 37.7%) [3.2 *]	24.8% (23.7%, 26.0%) [3.2 *]	34.4% (24.1%, 45.1%)
	outdoor	78.5% (77.0%, 79.8%) [3.7]	61.0% (58.7%, 63.3%) [3.5]	63.2% (61.0%, 65.4%) [2.9]	60.0% (58.9%, 61.1%) [2.7]	53.8% (52.5%, 55.1%) [3.6]	63.3% (53.0%, 79.3%)
Wild_Birds	indoor	5.4% (3.2%, 9.0%) [3.8 *]	3.4% (2.0%, 5.7%) [3.8 *]	4.3% (2.5%, 7.1%) [3.8 *]	4.2% (2.5%, 7.0%) [3.8 *]	2.7% (1.6%, 4.5%) [3.8 *]	4.0% (1.9%, 7.4%)
	outdoor	12.5% (7.6%, 20.3%) [3.8 *]	8.0% (4.7%, 13.3%) [3.8 *]	9.5% (5.7%, 15.6%) [3.5]	9.8% (5.9%, 16.1%) [3.8]	6.7% (4.0%, 11.1%) [4.0]	9.3% (4.7%, 17.0%)
Wild_Boar	indoor	19.1% (17.6%, 20.7%) [2.3 *]	12.4% (11.3%, 13.5%) [2.3 *]	15.4% (14.0%, 16.9%) [2.3 *]	15.1% (14.0%, 16.3%) [2.3 *]	9.9% (9.2%, 10.6%) [2.3 *]	14.4% (9.5%, 20.0%)
	outdoor	44.3% (42.1%, 46.7%) [2.8]	28.7% (26.7%, 30.6%) [2.5]	33.2% (30.8%, 35.6%) [2.3 *]	33.5% (31.8%, 35.3%) [2.4]	19.9% (18.7%, 21.1%) [2.0]	31.9% (19.2%, 45.7%)
Wild_Ruminants	indoor	22.0% (20.3%, 23.8%) [4.8 *]	14.3% (13.1%, 15.5%) [4.8 *]	17.7% (16.2%, 19.5%) [4.8 *]	17.4% (16.2%, 18.8%) [4.8 *]	11.4% (10.6%, 12.3%) [4.8 *]	16.6% (10.9%, 23.0%)
	outdoor	51.6% (49.0%, 54.3%) [5.8]	27.7% (25.9%, 29.5%) [4.2]	37.9% (35.3%, 40.6%) [4.8 *]	39.1% (37.1%, 41.1%) [5.1]	24.6% (23.2%, 26.1%) [4.6]	36.2% (23.7%, 53.2%)

## Data Availability

Appendix A for the reported results, including publicly archived datasets analyzed or generated during the study, can be found at GitHub, an online data repository, reachable through the following URL: https://github.com/arno314/Toxosources.

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
