# Peer review of "Systematic Review and Modelling of Age-Dependent Prevalence of Toxoplasma gondii in Livestock, Wildlife and Felids in Europe"

_pathogens, 2023, doi:10.3390/pathogens12010097_

Round 1
Reviewer 1 Report
The article “Systematic review and modelling of age-dependent prevalence of Toxoplasma gondii in livestock, wildlife and felids in Europe”, by Dámek et al., aims to estimate the prevalence of Toxoplasma in animal host species in Europe. The followed approach was a literature review and meta-analysis. The inclusion criteria are well defined and the data analysis is well conducted. However, in my point of view , this study has a limited interest.
Author Response
Dear reviewer,
We would like to thank you very much for your time spent reviewing our manuscript. We are sorry to hear that you found the study to be of limited interest. The model developed in the current study is already being used as a crucial building stone for a final output of a larger European One Health Project, and we believe that our work will find its place and prove valuable also to other studies.
Best regards,
Filip Dámek
Reviewer 2 Report
The manuscript “Systematic review and modelling of age-dependent prevalence of Toxoplasma gondii in livestock, wildlife and felids in Europe” by Dámek et al., presents the data of an extensive systematic literature review from 226 eligible publications of T. gondii seroprevalence and prevalence in Europe until June 2020. The study carried out a Bayesian approach for the estimation of the seroprevalence at the age most relevant for human infection by species. The model is of interest because allows to estimate data in areas of little or no data available to fill some data gaps. Although data on prevalence based on direct detection methods were included, the data were scarce in Europe, and they could not be included in the Bayesian hierarchical age-dependent model. Since I am not an expert on Bayesian models, I would not make any comments on the suitability of the model.
General comments:
The data for prevalence included in supplementary data (table S4) would better be included in the main text of the manuscript for easier visualization of those data.
In the supplementary excel files for species analysis, there is a column “Number of the article” which does not mean anything to the reader. The citation/reference of each of those studies instead would be useful for any reader of this manuscript for easier follow up of the literature included.
Specific comments:
Abstract: “In more than 50 animal species” or “in a selection of animal host species”. Please be more specific. Something like “feline species and farm animals and wildlife species intended for human consumption as indicated in line 110-111” or “livestock, wildlife and felids” as indicated in the tittle.
“The lowest seroprevalence in indoor-held lagomorphs from Easter Europe”. This result is somehow in contradiction of line 281 “the highest prevalence based on direct detection was in the brains of lagomorphs in the Western region” since prevalence really is indication of the presence of the parasite and not only contact with it.
The last sentence of the abstract is ambiguous “provide valuable inputs for future source attribution” and could be more specific and show the real achievements of the review and the model (e.g., useful to extrapolate data to areas without the needed data and therefore to fill data gaps….).
Lines 204-205. Cattle, buffaloes, and equids were excluded from the age-dependent seroprevalence modelling because literature indicates that there is no relationship between direct detection and seroprevalence. Was that relationship between observed in the studies included in the systematic review?
Line 283-284. “The lowest T. gondii prevalence was observed in the hearts of equids from the Eastern region (other tissues, Romania, n=82) at 0.0 % (95% CI: 0.0 – 0.4 %) (Fig. 2, Supplementary files – Table S4)”. However, in Table S4, other species and regions had also 0.0% prevalence: buffalo in East (n=74), equids in East (n-=82), lagomorphs in Southeast (n=52), pigs in Southwest (n=44) or wild birds in Southwest (n=5) as well. Please include in text.
The data on prevalence felids is of interest, being the definitive host. A very consistent low prevalence was observed in fecal samples in felids: 0.17% of 598 felids in North and 0.15% of 104309 in West. This information could be included in the text.
Seroprevalence
Line 321. Is the waiting time shown here (4 years on average to become seronegative) for all species included in the analysis?
Line 394 shows different data that those included in the abstract. The abstract would be more consistent showing the information included here.
Author Response
Dear reviewer,
We would like to thank you for your time spent with our manuscript and for providing much-appreciated feedback, thus improving the quality and readability of the manuscript. Please see the attachment for a point-by-point response to your questions and remarks.
Kind regards,
Filip Dámek
